# Preparation and Properties of Corn Starch/Chitin Composite Films Cross-Linked by Maleic Anhydride

**DOI:** 10.3390/polym12071606

**Published:** 2020-07-19

**Authors:** Peng Yin, Jinglong Liu, Wen Zhou, Panxin Li

**Affiliations:** 1College of Science, Nanjing Forestry University, Nanjing 210037, China; 13951778290@163.com (J.L.); wenzhounjfu@163.com (W.Z.); 2Agricultural and Forest Products Processing Academician Workstation, Luohe 462600, China; 15651851089@163.com; 3Post-Doctoral Research Center of Nanjiecun Group, Luohe 462600, China

**Keywords:** chitin, starch films, maleic anhydride, mechanical property, barrier property, antibiotic property

## Abstract

To improve the functional properties of starch-based films, chitin (CH) was prepared from shrimp shell powder and incorporated into corn starch (CS) matrix. Before blending, maleic anhydride (MA) was introduced as a cross-linker. Composite CS/MA-CH films were obtained by casting-evaporation approach. Mechanical property estimation showed that addition of 0–7 wt % MA-CH improved the tensile strength of starch films from 3.89 MPa to 9.32 MPa. Elongation at break of the films decreased with the addition of MA-CH, but the decrease was obviously reduced than previous studies. Morphology analysis revealed that MA-CH homogeneously dispersed in starch matrix and no cracks were found in the CS/MA-CH films. Incorporation of MA-CH decreased the water vapor permeability of starch films. The water uptake of the films was reduced when the dosage of MA-CH was below 5 wt %. Water contact angles of the starch films increased from 22° to 86° with 9 wt % MA-CH incorporation. Besides, the composite films showed better inhibition effect against *Escherichia coli* and *Staphylococcus aureus* than pure starch films.

## 1. Introduction

Starch films have been considered for many years as an alternative polymer material for petroleum-based plastic-related industries due to their inexpensive, renewable, and completely biodegradable properties as well as their potential barrier properties [1,2,3,4,5]. However, starch-based film has a number of disadvantages compared to synthetic plastics such as its brittleness, easy aging, and high hydrophilicity [6]. The absorption of water will accelerate the degradation and recrystallization of starch and lose its mechanical properties. At the same time, as an energy substrate, starch-based plastic will be susceptible to the action of microorganisms especially of fungus and then become useless [7,8,9,10]. 

Some additives were often used to overcome the above problems. The additives mainly included carboxylic acid modifiers [11], inorganic minerals (industrial calcium carbonate, biomass calcium carbonate, and nano-clay) [12,13,14], polysaccharide nanoparticles such as cellulose, starch, chitin and chitosan, alginate, etc. [15,16,17]. Reinforcing starch with polysaccharide nanoparticles is an appealing approach since both components are polar by nature and no compatibilization is required [18]. Chitin, a linear polysaccharide composed of (1–4)—linked 2-acetamido-2-deoxy-b-D-glucopyranose units, is the second prevalent form of polymerized carbon in nature. It is totally or partially deacetylated (often above 55%) products are named as chitosan. Chitin and chitosan possess remarkable properties—such as low density, large surface, hydrophilicity, chemical reactivity, and antimicrobial activity—and thus have been extensively studied for a broad variety of applications in bionanocomposites [19,20,21]. Chitin or chitosan are reported to be used as matrix or as nanofillers to prepare hydrogel or films that can be strategic for packing, adsorbent, and pharmaceutical applications [22,23,24,25]. 

The use of chitin or chitosan as nanofillers and their effect on the properties of starch films have been well estimated in researches. Properties seem to be dependent on the dosage and the size and shape of the nanofillers. Dosage is a general factor because any filler that exceeds the loading capacity of the matrix will bring about negative effect on the performance of the films. For example, in Qin et al.’s study [26], the tensile strength of maize starch films incorporated with 0.5–2% chitin nano-whiskers increased from 2.79 to 3.17 MPa but decreased to 2.37 MPa when the dosage of fillers were 5%. The size of chitin/chitosan fillers is another factor. Fillers with high molecular weight are favorable for the improvement of mechanical and barrier properties [27,28,29,30,31]. The aspect ratio of the fillers is another important factor in determining properties of the starch films. Salaberria et al. [27] investigated the role of chitin nanocrystals (CHNC) and nanofibers (CHNF) on properties in thermoplastic starch films. Superior data were found with the S/CHNF samples, in which the tensile strength jumped from 5 to 11 MPa in the starch nanocomposite films incorporated with 5 and 20 wt % of CHNF. However, as can be observed in the literatures, the increase of tensile strength was often accompanied by the sharp decrease of elongation at break in the composite films which indicated a more fragile behavior [27,29,30]. Further improving the properties of starch-based composites became essential.

Aiming to further improve the properties of starch films, thermoplastic starch nano-biocomposite films were prepared using chitin nanoparticles extracted from shrimp shells as fillers (S/CH) in this study. Maleic anhydrate (MA) was introduced before blending the two matrices and the obtained composite films were named as S/MA-CH. Films without maleic anhydrate incorporation named as S/CH were prepared as comparison. The mechanical, thermal, barrier, and antibacterial properties of the composite films were examined to assess the effect of MA-CH nanoparticles on the starch films. 

## 2. Materials and Methods

### 2.1. Materials

Normal maize starch (with an amylose content of approximately 32 ± 1%) was obtained from Zhucheng Xingmao Corn Development Co., Ltd. (Tengzhou, Shandong, China). The shrimp shell powder was purchased from Shandong Qilin animal husbandry (Linyi, Shandong, China). Glycerol was supplied by Tianjin Jiangtian Chemical Co. Ltd. (Tianjin, China). All other chemicals used in the present study were of analytical grade.

### 2.2. Preparation of Chitin Nanoparticles

Chitin was isolated from powder of mantis shrimp wastes according to Salaberria et al.’s [29] methods with some modification. Briefly, proteins were first removed using a 2 M NaOH solution at 25 °C for 24 h under vigorous stirring; then, minerals (CaCO_3_) were removed by 2 M HCl (37% *w*/*w*) solution at 25 °C for 3 h; Products from each step were washed with distilled water and filtrated. Pellets were dried at 60 °C for 12 h and then dispersed in 3 M HCl (37% *w*/*w*) at 100 °C for 3 h under reflux for acid hydrolysis, and finally, pigments were extracted using 2% KMnO_4_ solution at 50 °C for 4 h under vigorous stirring. The obtained products were filtered and washed with distilled water. The resulting chitin nanoparticles were dried at 60 °C overnight in an oven. 

### 2.3. Maleic Anhydride Modification and Film Preparation

The obtained chitin nanoparticles were washed and dried and then dispersed in absolute ethyl alcohol. 5% (w/w chitin) maleic anhydride was added to the mixed solution and maintained at 45 °C for 4 h under stirring. The solution was then adjusted to pH 7.0 and washed by ethyl alcohol and dried.

Maize starch/MA-chitin composite films (S/MA-CH) were prepared using a solution casting method. Briefly, 3.0 g of maize starch was dispersed in 100 mL of deionized water and aliquots of MA-CH (0, 1.0, 3.0, 5.0, 7.0, and 9.0 wt %, based on maize starch) were added to the starch suspension. The mixture was stirred for 30 min at 90 °C, then, 3.0 g of plasticizer (glycerol) were added and the suspensions were stirred for an additional 30 min at 90 °C Finally, the samples were cooled to room temperature and poured into a 10 × 10 cm^2^ acrylic mold and dried for 48 h at 25 ± 2 °C. Films with unmodified chitosan nanoparticles (S/CH) were also prepared as control. Sample codes were S, S/CH, S/MA-CH, respectively. All starch films were preserved in a relative humidity of 70 ± 2% chamber at 25 ± 2 °C for further testing. 

### 2.4. Fourier Transforms Infrared Spectroscopy

The absorbance spectra of the chitin nanoparticles, S/MA-CH composite films were recorded with an infrared spectrometer (VERTEX 70, Bruker, Hamburg, Germany) in attenuated total reflectance (ATR) mode. Powder samples were pressed by potassium bromide. IR spectra were measured at wavelengths from 400 cm^−1^ to 4000 cm^−1^. 

### 2.5. Tensile Measurements

The tensile tests were performed at room temperature in accordance with the ASTM D638 standard on a testing machine (SANS, MTS Systems Corporation, Shenzhen, China). Five to eight specimens were tested for each sample, and the average values of the measured properties were reported.

### 2.6. Contact Angle

Measurement of the contact angle was carried out at room temperature. The wetting behavior of the samples was measured and analyzed using a contact angle analyzer (DSA100, KRUSS, Hamburg, Germany) and a watered syringe. A drop of water was dropped on the sample, and its angle of incidence was measured soon after deposition using software. Each photo was taken for 0.016 s to 1 s.

### 2.7. Scanning Electron Microscope

The morphology of the samples was studied on a field emission scanning electron microscope (JSM-7600F, Takeno, Tokyo, Japan). The films were put into liquid nitrogen and its fractured cross-sections were used for SEM analysis. The fractured faces were vacuum coated with gold prior to analysis, and the tungsten filament was operated at 20 kV.

### 2.8. Differential Scanning Calorimetry and Thermogravimetric Analysis

DSC thermograms of a sample were recorded by a differential scanning calorimetry analyzer (DSC 204 F1, Netzsth, SELB, Bavaria, Germany). Samples were tested within a temperature range of 30–250 °C at a heating rate of 10 °C/min. Thermal stability curves of the samples were recorded on a thermogravimetric analyzer (TG 209 F1, Netzsch, SELB, Bavaria, Germany). The samples were analyzed under a nitrogen atmosphere over a temperature range of 25–600 °C at a heating rate of 20 °C/min.

### 2.9. Water Uptake

Films were cut into 20 × 20 mm specimens and dried at 105 °C for 2 h. After being weighed, the specimens were kept in a climate chamber (25 °C, RH 70%), then samples were taken out every 1 h and weighed until 6 h. The mass gain at each time (W, water uptake) was calculated as: W (%) = [(W_f_ − W_0_)/W_0_] × 100 where W_0_ is the sample’s initial mass and W_f_ is the sample’s mass after absorption time.

### 2.10. Water Vapor Permeability (WVP)

WVP was assessed using Dang et al.’s [30] methods with some modifications. Each film was cut into a round shape with a diameter of 22 mm and then placed on the open mouth of a test cup, which had an inner diameter of 20 mm and contained 8 g of dried calcium chloride. The films were then sealed to the cup using paraffin wax and placed in an incubator at 25 ± 2 °C and 70 ± 2% RH for 2 h for balance. Then, samples were weighed every 1 h until 6 h. The WVP was calculated as follows: WVP = WVTR/S(R1–R2) × X, where WVTR is the water transmission rate which was determined by the slope of the linear portion of a plot of weight gained versus time (g/h), R1 and R2 is the RH of incubator and test cup, respectively. X is the film thickness (m), S is the saturated water vapor pressure (Pa).

### 2.11. Antibacterial Activity

Antibacterial activities of starch films (control sample), and representative S/MA-CH composite films were examined as inhibitory effects against the growth of Gram-positive bacteria, *Staphylococcus aureus*, and Gram-negative bacteria, *Escherichia coli*. To study the antibacterial activities, changes in the growth of *S. aureus* and *E. coli* incubated in the broth medium were investigated following Qin et al.’s [26] methods. All strains from agar slant were aseptically inoculated in LB broth and subsequently incubated at 37 °C for 12 h under mild shaking, then, an inoculum (100 μL) of *S. aureus* and *E. coli* were aseptically transferred to 50 mL of LB broth containing film samples (2 × 2 cm^2^) and shaking at 37 °C for 12 h. The inhibitory effect was estimated by measuring the turbidity of the cultured medium at 600 nm using a spectrophotometer. The medium from shaking flask after 12 h cultivation was diluted 100,000× and transferred to agar plate and incubated at 37 °C for another 12 h, the colony growth on the plate were recorded.

## 3. Results and Discussion

### 3.1. Mechanical Properties

To investigate the effect of MA-CH on the mechanical properties of starch films, different percentages of MA-CH were incorporated into starch matrix to prepare S/MA-CH films. Starch films filled with the same concentration of CH were also prepared to make comparison. The tensile strength and elongation at break of all films were determined and the results were shown in Figure 1. Obviously, all starch films filled with chitin nanofillers became more resistant to the tensile. The tensile strength of films filled with 1–7 wt % MA-CH increased from 5.82 to 9.32 MPa. The pure starch film has a tensile strength of 3.89 MPa. As a comparison, the tensile strength of starch films with 7 wt % CH was 8.36 MPa. This is in accordance with the results of previous reports in which chitin or chitosan nanofillers were incorporated [31,32,33]. As described in the literature, the improvement of tensile strength was often attributed to the intermolecular hydrogen bonding between chitin and starch chains and the efficient stress transfer from the matrix to the chitin nano-size fillers. 

The slight decrease of tensile strength of the composite films with 9 wt % CH or MA-CH could be attributed to aggregation of the fillers which exceeded the loading capacity of starch matrix. This is in agreement with the results of Qin [26] and Chang [32] in which the tensile strength of the composite films decreased when the dosage of chitin nanoparticles exceeded 6 wt % and 1 wt % respectively. The loading capability of the starch matrix may be dependent on the size and shape of the fillers. The elongation at break of the starch films decreased with the CH incorporation, but remained 30% when the concentration of MA-CH was 7 wt % (the pure starch films is 87%) (Figure 1b). The superior mechanical property of the starch films incorporated with MA-CH nanoparticles was due to the cross-linking effect between chitin and starch by maleic anhydrate incorporation. This is also found in Wu et al.’s study in which citric acid was used as a crosslinker between starch and chitosan [34]. The tensile test curves were provided as Appendix A.

### 3.2. Reaction Mechanism

Figure 2 shows the ATR-FTIR spectra of chitin, starch films, and composite S/MA-CH films incorporated with different concentration of MA-CH. The starch films exhibited their characteristic bands including the 3276 cm^−1^ (O–H stretching), 2925 cm^−1^ (C–H stretching), 1651 cm^−1^ (bound water), 1148 cm^−1^ (C–O stretching) of starch which is correspond to the results of Lopez et al. (2014) [28] and Dan et al. (2016) [30]. The bands of chitin nanoparticles were also typical and were in agreement with those reported by Palpandi et al. (2009) [35] for α-chitin from shrimp. The bands at 3448 cm^−1^ corresponded to the stretching of N–H which was overlapped with the band of O–H stretching at 3363 cm^−1^ [36]. The band at 2930 cm^−1^ was attributed to the CH_3_ and CH_2_ stretching and the one located at 2857 cm^−1^ to CH bonds. The presence of amide I and II groups were evidenced by bands at 1663 and 1583 cm^−1^, respectively. Besides, other bands were identified at 1421 cm^−1^ (CH_3_ deformation and CH_2_ bending), 1088 and 892 cm^−1^ (ring stretching), 1030 cm^−1^ (CO). 

Incorporation of MA-CH did not significantly affect the spectra pattern of starch films, but some bands became more intense with the MA-CH incorporation. The strengthened bands mainly included 3426, 2926, 1146, 1643, 1416, and 1008 cm^−1^ and were due to the increment of some functional groups of both starch and MA-CH. For example, the more intensive and sharp band at 2926 and 1416 cm^−1^ was due to the increase of C–H bonds, and the intensities of the bands at 1643 and 1321 cm^−1^ gradually increased with the incorporation of MA-CH nanoparticles due to the presence of amine and carboxyl groups (N–H and C=O), respectively. Hydrogen bonding between starch and chitin molecules was expected but was difficult to evaluate from the spectra characterized at room temperature due to the effect of moisture [37]. Only a little shift of the band at 3276 cm^−1^ was observed in the spectrum indicative of hydrogen bonding formation between chitin and starch. The band at 1740 cm^−1^ (stretching of C=O) which was also increased with the increasing of MA-CH, indicated the increased crosslinking reaction between MA and polysaccharides. A similar phenomenon was also found in Wu et al.’s study in which they used citric acid as cross-linker in starch-chitosan composite films [34].

### 3.3. Morphological Characterization

Micrographs of the representative starch and starch-based composite films (S, S/CH3, S/CH7, S/MA-CH3, and S/MA-CH7) were presented in Figure 3. All images were taken from the fragile fractured surface of the films. It can be seen that the starch films without nanofillers have a uniform and smooth fracture surface, some cracks were observed which indicated a fragile property of the film. Incorporation of chitin nanoparticles in the starch matrix make the cross-sections look a little rough and a lot of wrinkles appeared. Meanwhile, cracks still appeared which indicated that incorporation of chitin nanoparticles did not improve the flexibility of starch films. When MA was introduced, the cross-section of the films became uniform and no cracks were observed. A scheme of the potential interactions between starch and MA-CH was also shown in the Figure. There are two kinds of probable intermolecular binding between the matrixes: (1) hydrogen bonding between the OH groups of starch backbone and the OH and residual NH_2_ groups at the surface of chitin; (2) covalent bonding between the carboxyl group of maleic anhydride and the OH groups of starch; (3) covalent bonding between the carboxyl group of maleic anhydride and the OH or residual NH_2_ groups of chitin nanofillers.

### 3.4. Thermal Analysis

DSC curves of starch films and S/MA-CH films were presented in Figure 4a. No glass transition or melting of polymers was observed in the temperature range analyzed. All films showed one single endothermic peak, corresponding to water evaporation [34]. The blend films showed a longer melting distance, which is in accordance with many other researches about starch films filled with nanofillers [26,31]. The maximum of the water evaporation peak decreased with the addition of 1–7 wt % MA-CH, but increased when the dosage of MA-CH is 9 wt %. This incident can be described in terms of intermolecular forces. When MA-CH was added, the hydrogen bonds between starch chains were destroyed and became weaker which made the water evaporation easier; However, more intermolecular bonds between starch and MA-CH were formed when the concentration of MA-CH reached 9 wt %. 

To further explore the thermal stability of the S/MA-CH films, thermogravimetric analysis was carried out and the results were shown in Figure 4b. All the films displayed two stages of thermal degradation. The first stage is the weight loss below 200 °C, which is mainly manifested as the vitalization of small molecules such as glycerol and water, as also shown in Figure 4. The second stage of weight loss in the temperature range 200–400 °C was attributed to dehydration of saccharide rings, depolymerization, and decomposition of the polymers [38]. The DTG curve in Figure 4b showed the maximum rate of degradation temperature of each film. It can be seen that with the addition of MA-CH, the maximum degradation temperatures decreased gradually from 332 °C to 312 °C, indicating a negative effect of MA-CH on the thermal stability of starch films. This may be due to: (1) the lower thermostability of MA-CH; (2) the destruction of crystalline structure of starch because of the reaction between the two carbohydrate polymers. A similar phenomenon was also observed in Salaberria et al.’s study in which they prepared starch composite films filled with chitin nanocrystals by melt-mixing or solution casting methods and all films filled with chitin nanocrystals showed a decreased thermostability [27,29]. 

### 3.5. Contact Angle

The water contact angle of the starch film and composite S/CH films with or without MA were measured and presented in Figure 5. All data were recorded soon after dropping. Starch films without nanofillers showed a contact angle of 22°. With the addition of CH nanofillers, the water contact angle decreased firstly, and then increased when the dosage of CH was above 5 wt %. The films with 9 wt % CH nanofillers have a contact angle of 57°. The decrease of the contact angles of the films at the beginning maybe due to the break of the intermolecular bonds of starch chains with nanofiller incorporation. Then, with the increment of the CH dosage, interaction between chitin and starch chains (e.g., hydrogen bonding) was formed, and the surface hydrophobicity of the S/CH films increased. When MA was introduced, cross-linking occurred between polymers, and hydrophobic groups such as C=O and C=C provided high water resistance for the films. As a result, obvious increment of water contact angle of the composite films was observed. This is in accordance with the study of Wu et al. (2019) [34] in which citric acid was introduced to potato starch/chitosan composite films. Formation of hydrophobic eater groups between MA and the polysaccharides leads to a decrease in the number of polar groups on the surface. In addition, it has been reported that the increased surface roughness will be helpful in improving the surface contact angle due to heterogeneous wetting [39]. The increased surface roughness of MA cross-linked S/CH films was also observed in the SEM images as shown in Figure 5. 

### 3.6. Water Vapor Permeability and Moisture Absorption

It is well known that the poor barrier property of starch films limited their applications. In this study, water vapor permeability (WVP) of the S/MA-CH films was determined to investigate the effect of MA-CH incorporation on the barrier property of starch films (shown in Figure 6a). It can be seen that the WVP value decreased sharply when 1 wt % MA-CH was incorporated and then increased slightly with the increasing of MA-CH dosage. However, all the blend films showed a lower WVP value compared with the pure starch films. The relative hydrophobicity of chitin as compared with that of starch and the cross-linking reaction between polymers may be the obstruction for water vapor transmission, and this was also confirmed by the higher water contact angle of TPS/MA-CH films than that of the neat starch film (Figure 5). This result was in agreement with Salaberria et al.’s [27] study, in which the WVP value of starch films decreased when 5 and 10 wt % chitin nanocrystal were added. Changes of the crystal structure of starch films and thus change the path of water vapor permeability because of the addition of chitin nanofillers may be the reason. Moreover, the replacement of hydrophilic hydroxyl groups with hydrophobic ester groups caused by cross-linker in this study is also the reason [40], and it can also explain the higher water contact angles of the S/MA-CH films than that of the S/CH films without maleic anhydride incorporation. 

The moisture absorption of the films was also investigated and shown in Figure 6b. The moisture absorption of starch films decreased when 1 wt % of MA-CH was added, then increased with the increasing concentration of MA-CH. The decrease of moisture absorption at the beginning may be due to the cross-linking between MA-CH and starch, which reduces the polarity and reduces the interaction between hydrophilic groups and water. Similar results were also found by [34] in which they used citric acid as crosslinker in potato starch/chitosan composite films. However, with the increasing of MA-CH dosage (above 7 wt %), the polar groups such as hydroxyl groups increased, and the water resistance reduced. 

### 3.7. Antibiotic Activity of Composite Films

All the films filled with chitin nanoparticles showed higher anti-mildew ability than the pure starch films (data shot shown). In this study, the antibacterial activities of the starch and S/MA-CH composite films were estimated. *S. aureus* and *E. coli* were selected for testing which were familiar Gram-positive bacteria and Gram-negative bacteria, respectively. The optical density (OD, absorbance at 600 nm) of culture medium including the starch or composite starch films was measured after inoculated by the microorganisms. Composite starch films filled with 5 wt % and 7 wt % of MA-CH were chosen for estimation and the results were shown in Figure 7a. The corresponding colony growth on agar plate was shown in Figure 7b. Since the ODs of the medium increased with the growth of microorganisms, lower absorbance at 600 nm indicated higher antibacterial activity of the test material [26]. Obviously, both of the growth of *S. aureus* and *E. coli* was suppressed in the suspensions contained S/MA-CH films. It is in agreement with Lopez et al.’s study [28]. Films with 5 wt % CH or MA-CH concentration showed superior antibacterial activities than that with 7 wt % nanofillers. As depicted in literature, the antimicrobial mechanism of the films could be due to the interactions between positively charged chitin and negatively charged bacterial cell membranes, which results in increased membrane permeability and eventually causes the rupture and leakage of the material [41]. In this study, no obvious differences were observed in the antimicrobial activities of the S/CH composite films against *E. coli* and *S. aureus*. The limitation of antibacterial activity may be due to the weak migration of chitin from the film to the broth, and the higher surface hydrophobic property of the S/MA-CH films which impede the adhesion of microorganism may be another factor [42]. 

## 4. Conclusions

Thermoplastic starch-based nano-biocomposite films with superior mechanical property, higher surface hydrophobicity, and barrier properties were successfully prepared by introducing chitin nanoparticles as fillers and maleic anhydride as cross-linker. Introduction of chitin nanoparticles and their good dispersion in thermoplastic starch matrix were confirmed by ATR-FTIR and FE-SEM analysis. 

The properties improvement was due not only to the rigid and hydrophobic properties of chitin nanoparticles which was derived from shrimp shell wastes, but also to the strong adhesion between starch and chitin and cross-linking reaction via maleic anhydride. The mechanical and barrier properties of the nano-biocomposites are related to the nanofillers load. The films cross-linked by maleic anhydride especially displayed better mechanical properties and higher water contact angles than the composite films without cross-linker. The films with MA-CH incorporation showed better antibacterial capabilities than the pure starch films, and thus have potential application for packaging or products requiring delayed biodegradation. 

## Figures and Tables

**Figure 1 polymers-12-01606-f001:**
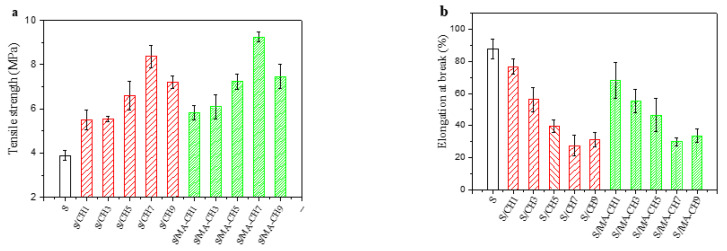
Tensile strength (**a**) and elongation at break (**b**) of S films, S/CH 1% films, S/CH 3% films, S/CH 5% films, S/CH 7% films, S/CH 9%, S/MA-CH 1% films, S/MA-CH 3% films, S/MA-CH 5% films, S/MA-CH 7% films, and S/MA-CH 9% films.

**Figure 2 polymers-12-01606-f002:**
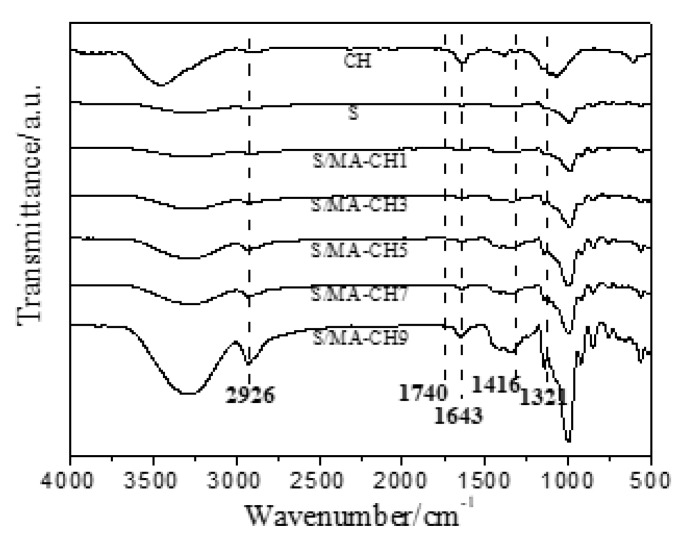
ATR-FTIR spectra of chitin nanoparticle (CH), starch films (S), S/CH 1% films, S/CH 3% films, S/CH 5% films, S/CH 7% films, S/CH 9%, S/MA-CH 1% films, S/MA-CH 3% films, S/MA-CH 5% films, S/MA-CH 7% films, and S/MA-CH 9% films.

**Figure 3 polymers-12-01606-f003:**
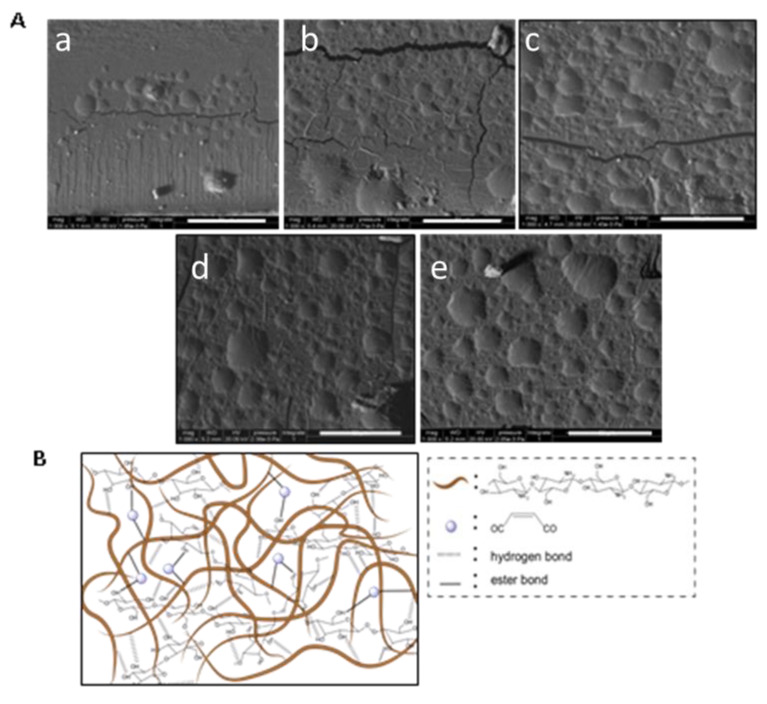
**A**. SEM photographs (1000× amplification) of starch film (**a**), S/CH 3% film (**b**), S/CH 7% film (**c**), S/MA-CH 3% film (**d**), S/MA-CH 7% film (**e**); **B**. illustrative scheme of S/MA-CH films. Scale bars indicate 50 μm.

**Figure 4 polymers-12-01606-f004:**
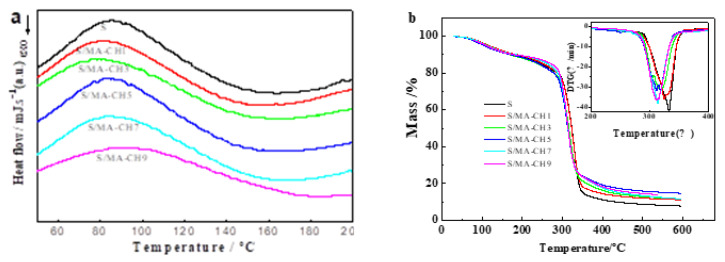
DSC thermograms (**a**) and thermogravimetric (TGA) and derivative thermograms (DTG) (**b**) of S films, S/CH 1% films, S/CH 3% films, S/CH 5% films, S/CH 7% films, S/CH 9%, S/MA-CH 1% films, S/MA-CH 3% films, S/MA-CH 5% films, S/MA-CH 7% films, and S/MA-CH 9% films.

**Figure 5 polymers-12-01606-f005:**
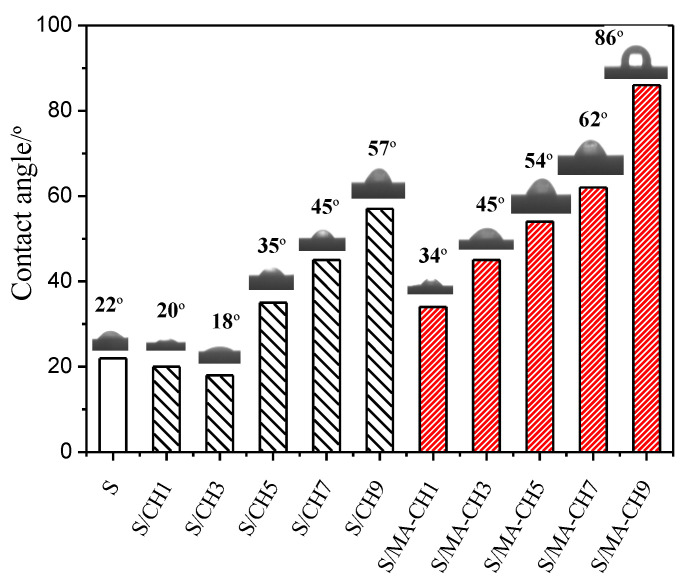
Water contact angle of S films, S/CH 1% films, S/CH 3% films, S/CH 5% films, S/CH 7% films, S/CH 9%, S/MA-CH 1% films, S/MA-CH 3% films, S/MA-CH 5% films, S/MA-CH 7% films, and S/MA-CH 9% films.

**Figure 6 polymers-12-01606-f006:**
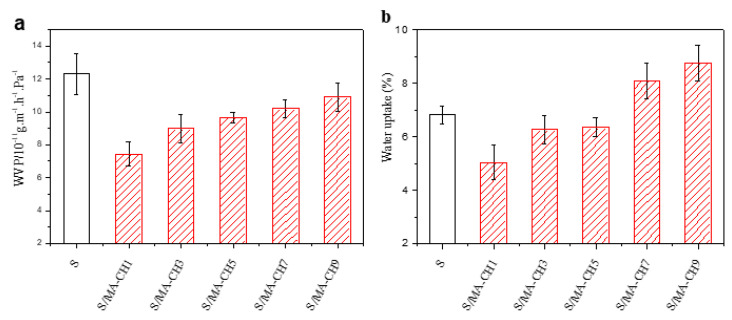
Water vapor transmission rate (**a**) and water uptake (**b**) of S films, S/CH 1% films, S/CH 3% films, S/CH 5% films, S/CH 7% films, S/CH 9%, S/MA-CH 1% films, S/MA-CH 3% films, S/MA-CH 5% films, S/MA-CH 7% films, and S/MA-CH 9% films.

**Figure 7 polymers-12-01606-f007:**
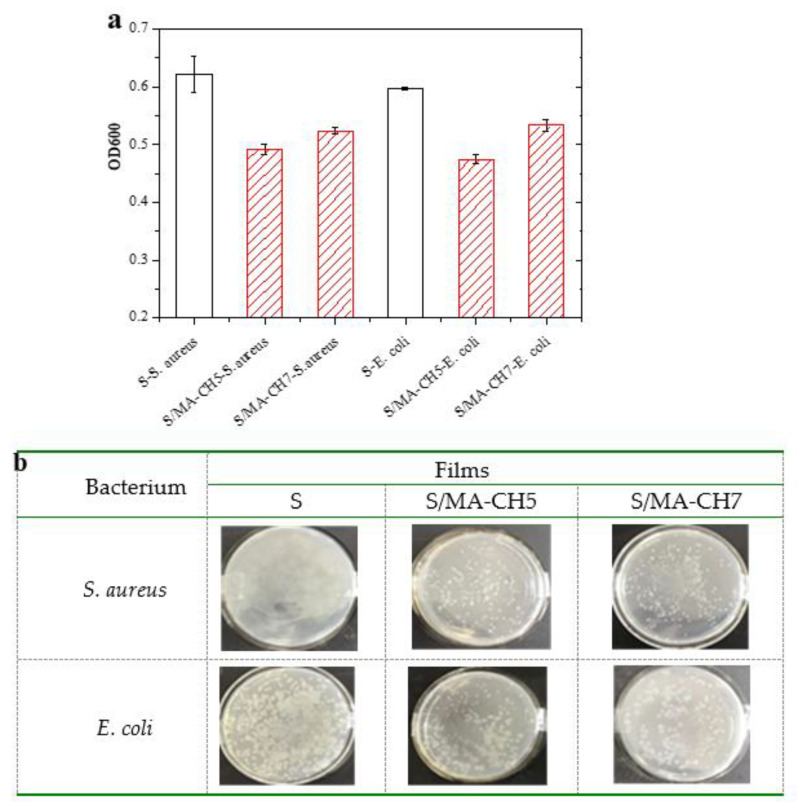
Inhibitory effect (**a**) of S films, S/CH 5% films, S/CH 7% films, S/MA-CH 5% films and S/MA-CH 7% films against *S. aureus* and *E. coli*, and pictures of agar plates (**b**) inoculated by 100 μL of diluted broth from the corresponding flask. Pictures were taken after 24 h of cultivation.

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
