# Peer review of "Preparation and Properties of Corn Starch/Chitin Composite Films Cross-Linked by Maleic Anhydride"

_polymers, 2020, doi:10.3390/polym12071606_

Round 1

Reviewer 1 Report

by and large, it is a well done work, easy to understand and the results are well justified according to the methods employed. However, it's hard for me to believe that maleic anhydride is the crosslinking agent, since it is well known that this compound hydrolyzes fully and very quickly in an aqueous medium. I recommend checking: "Maleic Anhydride#. Authors: Trivedi, B and and make that change in the entire document. Regarding to Figure 3, it should be revised becuse the crosslinking agent presented does not correspond to either anhydride or maleic acid

Finally, I consider that the discussion can be improved by comparing the contact angle data together with that of water vapor transmission.

Reviewer 2 Report

In this manuscript, Yin et al., have discussed the preparation and properties of starch/chitin composite films along with the adhesion of gram positive and negative bacteria. The study is interesting and I recommend it for publication, once the following comments have been well taken care of:

  1. How many samples have been tested for each case in Figure 1 ?
  2. How standard deviation has been calculated ? What is the confidence interval ?
  3. I suggest authors to add tensile test curves associated with figure 1 in supplementary. Tensile test curves will provide more information about the deformation behavior of the composite films. 
  4. Scale bars in the SEM images in figure 3 are not clearly visible. 
  5. Authors should include this relevant article on the adhesion of bacterias over the films (ACS Appl. Mater. Interfaces 2017, 9, 23, 19371–19379).

Reviewer 3 Report

The manuscript Preparation and properties of corn starch/chitin composite films cross-linked by maleic anhydride reported the preparation of starch-based films modified with chitin and crosslinked with maleic anhydride in order to improve the functional properties of the film. They carried out a systematic study of different compositions and tested the mechanical properties, morphological characteristics, and antibiotic activity of the films. The research design is adequate and systematic, the study is very well conducted and the results are clearly presented. But unfortunately, Wu et al (https://doi.org/10.1016/j.foodhyd.2019.105208) have reported a very similar paper taking out the novelty of the work. It would be nice if they introduce a sentence pointing in which form their research is original.

Reviewer 4 Report

The work is interesting and results well presented. I suggest some revisions before publication.
- Figure 3 Scale bars for SEM are not clearly visible.
- TGA, fig 4 b. “weight loss” should be changed in “mass” or “”weight” as it starts from 100 %.
- “The DTG curve in Figure 4b showed the maximum decomposition temperature of each film”. Actually the curves are DTG and therefore they show the temperature corresponding to the “maximum rate of degradation”.
- Conclusion section needs revision to be more clear. In the present form it is a list of results by each method. A more comprehensive point of view is recommended. The core findings, novelty and perspectives open by this work should be stated here.
- Contact angle: Are the reported values extrapolated at t=0, equilibrium (if any) or just after deposition?
- References might be updated considering recent findings in bionanocomposite materials based on chitosan (see for instance: Nanomaterials 2020, 10(6), 1194; https://doi.org/10.3390/nano10061194; Coatings 2019, 9(2), 70; https://doi.org/10.3390/coatings9020070; Materials 2020, 13(3), 688; https://doi.org/10.3390/ma13030688; New J. Chem., 2018,42, 8384-8390, https://doi.org/10.1039/C8NJ01161C )

Round 2

Reviewer 4 Report

Revised ms is suitable for publication